# Pairwise Explanations: Towards A Novel Task-agnostic Paradigm in Explainable Artificial Intelligence

## Abstract

In this study, we introduce Pairwise-IBISA (P-IBISA), a novel extension of the Information Bottleneck with Input Sampling for Attribution (IBISA) method. Unlike traditional approaches, P-IBISA generates explanations directly from encoder representations, eliminating the need for task-specific logits. This design enables interpretability across a wide range of applications, including image retrieval and vision–language grounding. It is compatible with models trained for classification as well as those pre-trained using self-supervised learning strategies. P-IBISA operates by computing a mask over the input image using a pairwise loss that aligns the embeddings of the masked image with a target embedding. This target can be derived from another image, the image itself, or a different modality—such as text in models like CLIP. We conducted a quantitative evaluation of P-IBISA on models designed for three distinct tasks: image classification, vision–language grounding, and image retrieval. Across these tasks, P-IBISA consistently demonstrated superior or competitive performance compared to state-of-the-art methods, despite being task- and model-agnostic. In particular, for visual-language grounding, we surpass the current state-of-the-art on the Confidence Increase (by at least 0.300 points) and Confidence Drop (by at least 12 points) metrics across multiple datasets. Qualitative analysis further reveals that P-IBISA produces sharper and semantically richer saliency maps, effectively highlighting meaningful features in both CNNs and ViTs pre-trained on unlabeled data. By decoupling explanations from final outputs, P-IBISA advances the field of xAI beyond task-specific evaluation, offering a unified framework for attribution across diverse scenarios.

## 1 Introduction

Explainable artificial intelligence (xAI) has emerged as a vital discipline, striving for human-interpretable insights into model behavior without compromising predictive accuracy (Rio-Torto et al., 2020). Beyond building trust, xAI facilitates regulatory compliance, aids in debugging errors, and ensures the ethical deployment of AI in sensitive contexts (Brás et al., 2025).

In computer vision, xAI primarily centers on attribution methods, which aim to identify the input image regions most influential to a model's predictions (Müller, 2024). These techniques are broadly classified into two categories: non-agnostic methods, which leverage the model's internal structure (model-dependent), and agnostic methods, which operate independently of the model's architecture.

Coelho & Cardoso (2025) recently introduced the Information Bottleneck with Input Sampling for Attribution (IBISA), a state-of-the-art (SOTA) computationally efficient, model-agnostic method for generating saliency maps for classification models. In this study, we advance this line of research by proposing P-IBISA (Pairwise-IBISA), a novel xAI paradigm that goes towards application-agnostic explainability based on pairwise explanations (see Figure 1). P-IBISA generates explanations using only the model encoder, rendering it decoder-agnostic and independent of output logits.

Decoupling explanations from application-specific outputs enables P-IBISA to tackle a broader range of interpretability questions, such as *"Why was this image encoded this way?"*, *"What makes these two images similar?"* or *"What aspects of this image align with the given text?"*. This flexibility allows for unprecedented insight into complex systems across diverse contexts.

In summary, this work makes the following contributions:

1. A novel decoder-agnostic xAI technique for saliency map generation;

2. A new approach to qualitatively evaluate models trained using self-supervised strategies with only unlabeled data;

3. The use of P-IBISA as a method to generate explanations for visual-language (CLIP) and image retrieval models;

4. A comparative analysis of P-IBISA against SOTA xAI methods on different tasks.

The remainder of this paper is structured as follows: Section 2 surveys the existing xAI methods for saliency map generation; Section 3 describes the P-IBISA methodology; Section 6 presents experimental results and comparisons; and Section 7 concludes with findings and future directions. The implementation code is publicly available in a GitHub repository[1] to support reproducibility.

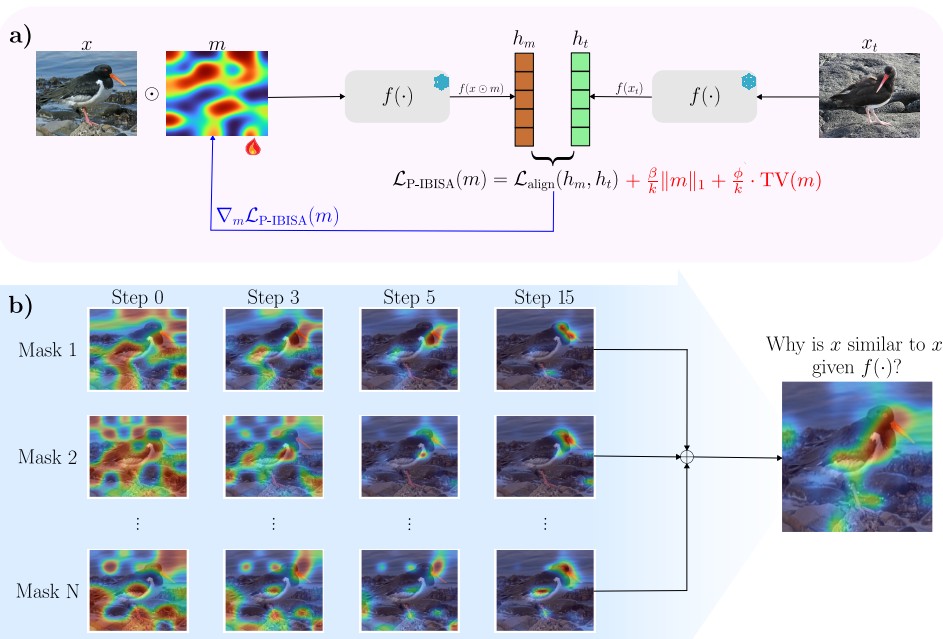

Figure 1: **P-IBISA** operates by finding what regions in an image $x$ are similar to a target image $x_t$ given an encoder $f(\cdot)$. **a)** A mask $m$ placed over $x$ is optimized to align the embeddings obtained from the masked input $h_m = f(x \odot m)$ with the ones obtained from a target image $h_t = f(x_t)$. During optimization, the mask $m$ is regularized to be constrained to a local region by penalizing its norm $\|m\|_1$ and its total variation $TV(m)$, while the $\beta$ and $\phi$ parameters control how sparse the final saliency is. **b)** A number $N$ of masks is sampled and optimized, with the final saliency being computed as their average. The method identifies the similarities between two images, as presented in the figure, but it is not limited to it. The target embedding $h_t$ may be obtained from $x$ itself, by making $x_t = x$, or even from the encoding of other modalities, such as text in visual language models.

## 2    RELATED WORKS

Among non-agnostic approaches, SOTA techniques often extend the foundational concepts of Grad-CAM (Selvaraju et al., 2017). These methods rely on access to the models' internal layers to compute derivatives that quantify the sensitivity of a model's decision to specific input regions (Chattopadhay et al., 2018). Although straightforward, derivative-based methods can suffer from limitations such as gradient saturation and susceptibility to noise, which can potentially yield inconsistent or unreliable explanations.

---

[1] https://github.com/future_repo_PIBISA

Methods based on the Information Bottleneck Principle (IBP) (Tishby et al., 2000), such as Information Bottleneck for Attribution (IBA) (Schulz et al., 2020) and Comprehensive Information Bottleneck for Attribution (CoIBA) (Hong et al., 2025), overcame the issues of Grad-CAM, producing more accurate and interpretable explanations. However, these methods still have model-specific constraints, which render them impractical when internal structures are unavailable.

To address these limitations, model-agnostic methods (e.g, LIME (Ribeiro et al., 2016), RISE (Petsiuk et al., 2018)) focus solely on the model's input-output relationship, offering explanations by analyzing changes in the input and corresponding shifts in output logits. In practice, this requires simulating a wide range of input masking scenarios to capture fine-grained feature importance, resulting in high computational costs.

More recently, Coelho & Cardoso (2025) introduced IBISA, transforming the baseline IBA into a model-agnostic method by shifting the bottleneck from the model's internal architecture to the input level, enabling the computation of saliency maps for any classification model that one can back-propagate through. This approach reduced the computational costs associated with other model-agnostic methodologies, while preserving performance.

A new challenge that is drawing attention from the xAI community is the explainability of visual-language models. Zhu et al. (2025) and Wang et al. (2023) introduced NIB and M2IB, leveraging the IBP to generate saliency maps that highlight the image regions related to a given text input in CLIP (Radford et al., 2021), improving the interpretability of this widely used architecture. Nevertheless, these methods are still non-agnostic, being only publicly available for CLIP with a ViT-B/32 backbone.

In this spirit, our current study extends IBISA (Coelho & Cardoso, 2025) by transforming the model-agnostic bottleneck into a decoder-agnostic approach, where we can directly interpret the encoder independent of output logits, enabling us to generate explanations for CNNs and ViTs trained with different self-supervised or fully-supervised strategies. P-IBISA also sets the state-of-the-art in the visual-language grounding task, where NIB and M2IB are surpassed in all metrics (see Tables 2 and 3).

## 3 PROPOSED METHOD

To provide a clear and structured overview, the proposed method is organized into three key components: overview and motivation, mask generation, and mask optimization.

### 3.1 OVERVIEW AND OBJECTIVE

The goal of P-IBISA is to generate saliency masks that reveal the most semantically relevant regions of an input image, as determined by a target embedding. The masks are optimized such that, when applied to the image, the resulting masked input yields an embedding close to a reference target—either the original image's own embedding or one from a different modality (e.g., text in vision-language models).

Formally, let $f : \mathbb{R}^{C \times W \times H} \to \mathbb{R}^d$ denote an encoder that maps an image to a $d$-dimensional embedding space, and let $h_t \in \mathbb{R}^d$ denote a target embedding obtained from a target input $x_t$. The objective is to determine a mask $m \in [0, 1]^{W \times H}$, such that the distance between $f(x \odot m)$ and $h_t$ is minimized as expressed in Eq. 1:

$$m^* = \underset{m \in [0,1]^{W \times H}}{\arg \min} \; \mathcal{D}(f(x \odot m), h_t), \tag{1}$$

where $\mathcal{D}(\cdot, \cdot)$ denotes a dissimilarity or distance function in the embedding space and $x \odot m$ is the element-wise multiplication between the mask and every channel of the input image.

P-IBISA optimizes multiple masks from different random initializations to ensure robustness and capture diverse semantically relevant regions. These optimized masks are aggregated by averaging to produce a final saliency map. Given $N$ optimized masks $\{m_1, m_2, \ldots, m_N\}$, where each $m_n \in [0, 1]^{W \times H}$, the saliency map $s$ is computed as Eq. 2:

$$s = \frac{1}{N} \sum_{n=1}^{N} m_n, \tag{2}$$

where $s \in [0, 1]^{W \times H}$ represents the final saliency map, with higher values indicating the regions in $x$ that are most similar to $x_t$, given an encoder $f$. The averaging step enhances explanation stability by mitigating the effects of local optima. For a more straightforward understanding of the procedure, we provide an overview of the P-IBISA algorithm in Algorithm 1:

---

**Algorithm 1** P-IBISA Saliency Map Generation

---

1: **Input**: Image $x \in \mathbb{R}^{C \times W \times H}$, encoder $f : \mathbb{R}^{C \times W \times H} \to \mathbb{R}^d$, target embedding $h_t \in \mathbb{R}^d$, number of masks $N$, grid size rate $l$, hyperparameters $\alpha, \beta, \phi$, iterations $T$, learning rate $\eta$.
2: **Output**: Saliency map $s \in [0, 1]^{W \times H}$.
3: Initialize empty list $M$ to store optimized masks.
4: **for** $n = 1$ to $N$ **do**
5:     Sample low-resolution grid $g \in \mathbb{R}^{w \times h}$, where $w = W/l$, $h = H/l$, and $g_{i,j} \sim \mathcal{U}(-\alpha, \alpha)$.
6:     Upsample $g$ to $m' \in \mathbb{R}^{W \times H}$ using bicubic interpolation.
7:     Compute initial mask $m_{i,j} = \sigma(m'_{i,j})$, where $m \in [0, 1]^{W \times H}$.
8:     **for** $t = 1$ to $T$ **do**
9:         Compute embedding $f(x \odot m)$.
10:        Compute alignment loss $\mathcal{L}_{\text{align}}(f(x \odot m), h_t)$.
11:        Compute regularization terms for sparsity and smoothness.
12:        Compute total loss $\mathcal{L}_{\text{P-IBISA}}(m)$.
13:        Update $m$ using Adam optimizer with learning rate $\eta$.
14:     **end for**
15:     Append optimized mask $m$ to $M$.
16: **end for**
17: Compute final saliency map $s_{i,j} = \frac{1}{N} \sum_{m \in M} m_{i,j}$.
18: **Return** $s$.

---

Further details about the mask generation and optimization processes, along with the associated loss functions, are presented in the following sections.

### 3.2 MASK INITIALIZATION

For an input image $x \in \mathbb{R}^{C \times W \times H}$, we generate an initial mask $m \in [0, 1]^{W \times H}$ following the methodology of IBISA (Coelho & Cardoso, 2025). The initialization begins by sampling a low-resolution grid $g \in \mathbb{R}^{w \times h}$, where $w = W/l$ and $h = H/l$ (typically $l = 32$), from a uniform distribution as in Eq. 3:

$$g_{i,j} \sim \mathcal{U}(-\alpha, \alpha), \tag{3}$$

where $\alpha > 0$ controls the sampling range, ensuring varied initializations. This grid is up-sampled to the full input resolution $\mathbb{R}^{W \times H}$ using bi-cubic interpolation, producing an intermediate mask $m'$. To ensure the mask remains bounded and differentiable, we apply a sigmoid transformation given in Eq. 4:

$$m_{i,j} = \sigma(m'_{i,j}) = \frac{1}{1 + \exp(-m'_{i,j})}, \quad m \in [0, 1]^{W \times H}. \tag{4}$$

This approach yields a smooth, gradient-friendly mask suitable for optimization. Bi-cubic interpolation promotes spatial coherence, reducing abrupt transitions, while random sampling of the grid encourages diverse exploration across multiple optimization runs, enhancing the robustness of the resulting saliency maps.

### 3.3 MASK OPTIMIZATION

After generating the initial mask, P-IBISA projects the masked input into the embedding space to align it with a target embedding. Let $f : \mathbb{R}^{C \times W \times H} \to \mathbb{R}^d$ be an encoder mapping an input image $x$ to a $d$-dimensional embedding space. Given a target embedding $h_t \in \mathbb{R}^d$—which may represent the embedding of the unmasked input $f(x)$, another image, or a different modality (e.g., text in vision-language models)—the goal is to optimize a mask $m \in [0, 1]^{W \times H}$ such that the embedding of the masked input, $f(x \odot m)$, closely approximates $h_t$.

We model this task using the Information Bottleneck Principle, as formalized in Eq. 5:

$$\hat{m} = \min_m \left( I[x; (x \odot m)] - I[h_t; f(x \odot m)] \right) \tag{5}$$

where we seek to find the mask $\hat{m}$ that minimizes the mutual information between the original image and its masked version, while preserving the relevant information on the input to reconstruct the target embedding $h_t$.

To achieve this objective, we used a modified version of the loss function introduced by IBISA, which has been shown to indirectly solve the IBP in this setup, presented in Eq. 6:

$$\mathcal{L}_{\text{P-IBISA}}(m) = \mathcal{L}_{\text{align}}(f(x \odot m), h_t) + \frac{\beta}{k}\|m\|_1 + \frac{\phi}{k} \cdot \text{TV}(m), \tag{6}$$

in which the first term $\mathcal{L}_{\text{align}}(\cdot, \cdot)$ is the cosine embedding loss used to maximize $I[h_t; f(x \odot m)]$, as in M2IB (Wang et al., 2023) and NIB (Zhu et al., 2025). The second term $\|m\|_1 = \sum_{i,j} |m_{i,j}|$ enforces sparsity, and the third term $\text{TV}(m)$ promotes spatial coherence via total variation regularization as denoted in Eq. 7.

$$\text{TV}(m) = \sum_{i,j} \left( |m_{i,j} - m_{i-1,j}| + |m_{i,j} - m_{i,j-1}| \right), \tag{7}$$

In Eq. 6, the constant $k = C \times W \times H$ serves as a normalization for the mask complexity terms. In contrast to IBISA, which optimizes masks using a cross-entropy loss tied to model predictions, P-IBISA operates solely in the embedding space, using $\mathcal{L}_{\text{align}}$, which makes it decoder-agnostic, as it eliminates the need to access output logits.

The $\beta$ and $\phi$ parameters, which control the sparsity of the final attribution, are defined a priori by the user in IBISA. In this work, we introduce a framework that treats $\beta$ and $\phi$ as learnable parameters during mask optimization. To accommodate this new objective, we adapt Eq. 6 following the strategy described by (Kendall et al., 2018), resulting in the final loss function presented in Eq. 8:

$$\mathcal{L}_{\text{P-IBISA}}(m) = \mathcal{L}_{\text{align}}(f(x \odot m), h_t) + e^{-\beta}\|m\|_1 + e^{-\phi} \cdot \text{TV}(m) + \frac{\beta + \phi}{t} + e^{-t}, \tag{8}$$

where we introduce the new $t$ parameter to prevent $e^{-\beta}$ and $e^{-\phi}$ from reaching values that over-penalize mask complexity, which would result in explanations that are too sparse.

The P-IBISA loss function, as Eq. 8 indicates, can produce negative values through the interplay of learnable $\beta$ and $\phi$ parameters, which dynamically weight regularization terms against the alignment objective. This process does not pose problems for mask optimization in our application, as it encourages sparse, high-confidence attributions by rewarding precise alignments over conservative penalties, thereby enhancing robustness (Terven et al., 2025). Moreover, the resulting negative losses provide a stable gradient signal that promotes efficient convergence without requiring non-negativity constraints, supporting adaptive sparsity control in xAI frameworks (Wu et al., 2024; Ye et al., 2024).

With this in mind, as we introduce the new strategy, it is important to note that P-IBISA inherits from IBISA its low computational cost and its ability to control the sparsity of the attribution. Specifically, the $\beta$ and $\phi$ values can be set directly by the user, rather than being learned during the mask optimization process (see Appendix E).

In terms of optimization, a standard gradient descent algorithm (e.g., Adam) with backpropagation is sufficient, as all operations, including upsampling and sigmoid normalization, are differentiable. Furthermore, we employ multiple random initializations to obtain a set of masks optimized for different local minima, which are then averaged to compute the final attribution. For more detailed information, we would like to reiterate that the full implementation is available in the project repository.

## 4 EVALUATION METRICS

We evaluate the performance of P-IBISA on three different tasks: image classification, text-image alignment, and image retrieval. To evaluate P-IBISA for classification, we first generate the saliency maps and then compute the commonly used *MoRF* and *LeRF* metrics (Petsiuk et al., 2018). For text-image alignment, we computed the *Confidence Increase* (CI) and the *Confidence Drop* (CD)

metrics introduced by (Chattopadhay et al., 2018), and adopted in previous works (Zhu et al., 2025; Wang et al., 2023) that addressed this task.

We also introduced two new metrics, Cap MoRF and Cap LeRF, which are adapted versions of MoRF and LeRF for the captioning problem. We first applied BLIP (Li et al., 2022) to generate captions for 1000 images sampled from the MS-COCO 2017 validation set (Lin et al., 2014), which were later used as targets to compute attributions for CLIP. We then compute the BLEU (Papineni et al., 2002), METEOR (Banerjee & Lavie, 2005), and ROUGE (Lin, 2004) scores between the original caption and the caption resulting from degraded versions of the image input, following the same procedure used to compute MoRF and LeRF.

To evaluate the retrieval task, we again used the CI and CD, where we replaced the observed output with the similarity between the query embeddings and the retrieved images. Since the model is trained to retrieve images based on Euclidean distance (see the next Section), we used the inverse of the Euclidean distance to measure similarity between embeddings.

## 5 EXPERIMENTAL SETUP

The masks are initialized with $w = W/32$, $h = H/32$, and $\alpha = 3$. To compute the final saliency map, we average 20 different masks optimized over 15 iterations using the Adam optimizer with a learning rate of 1. Unless otherwise specified, we used the setup where the $\beta$ and $\phi$ parameters are learned during the mask optimization procedure, and the cosine embedding loss serves as the alignment function. The learnable parameters are initialized with $\beta = \phi = 0$ and $t = 1$.

We qualitatively compared the attributions generated for the ResNET-50 and ViT-B/16 architectures when trained using the self-supervised strategies: DINOv1 (Caron et al., 2021), DINOv2 (Oquab et al., 2023), Barlow Twins (Zbontar et al., 2021), SwAV (Caron et al., 2020), and iBOT (Zhou et al., 2021) to evaluate if these models extract human-relatable features from the input. In these cases, the target embedding $h_t$ is the input vector for the classifier head.

For the classification task, we evaluated P-IBISA for CNNs and ViTs on 10,000 images from the ImageNet validation set (Deng et al., 2009). For CNNs, we compared our approach against the SOTA methods Grad-CAM (Selvaraju et al., 2017), RISE (Petsiuk et al., 2018), IBA (Schulz et al., 2020), and IBISA (Coelho & Cardoso, 2025) on the VGG-16 and ResNET-50 models. We also evaluated P-IBISA on the ViT-B/16 model, where we compared our technique with the Attention Rollout (Abnar & Zuidema, 2020), Guided Attention (Leem & Seo, 2024), and Chefer (Chefer et al., 2021), which are methods designed to generate saliencies for ViTs.

The MoRF and LeRF metrics were computed using $8 \times 8$ pixel blocks, which were removed from the input by turning their values to zero. For the visual-language task, we evaluated the metrics for the CLIP (Radford et al., 2021) with a ViT-B/32 backbone. We compared the performance of P-IBISA against the previous SOTA methods RISE (Petsiuk et al., 2018), M2IB (Wang et al., 2023) and NIB (Zhu et al., 2025) on the validation sets of MS-COCO 2017, Flickr8k (Cui et al., 2024) and MS-CXR (Boecking et al., 2022) datasets.

To assess the performance of P-IBISA in a domain-specific scenario, we evaluated our method on a medical image retrieval system developed to support the prediction of aesthetic outcomes in breast cancer (BrCa) patients following locoregional treatments (Zolfagharnasab et al., 2024). The dataset is private, and consists of 2,193 images of patients' upper torsos, each paired with clinical annotations identifying the 10–15 most similar cases per query, based on expert evaluations of aesthetic outcomes using the Harris scale (Rose et al., 1989).

The curation process yielded over 150 non-overlapping catalogues, with each query image linked to a ranked set of clinically similar cases. The dataset was collected within the scope of the XXXXXX project, adhering to all required ethical guidelines, and has been previously used in other publications. The retrieval model is based on a BeiT backbone (Bao et al., 2021), and we compare P-IBISA against the Integrated Gradients (Sundararajan et al., 2017) and SBMS (Dong et al., 2019) methods. Since the model trained for this task retrieves the images based on the Euclidean Distance, we needed to replace the cosine similarity in Eq. 6 by the mean squared error.

# 6 RESULTS

Figure 2 presents the saliency maps generated for the ResNET-50 and ViT-B models when the target embedding is obtained from the image itself. We used P-IBISA to evaluate the semantics of the features extracted by encoders trained with labeled data (fully supervised) and with only unlabeled data (self-supervised). By analyzing the attention values, the authors of iBOT (Zhou et al., 2021) and DINOv1 (Caron et al., 2021) showed that ViT architectures can learn semantically relevant features when pre-trained with only unlabeled data. Nevertheless, their analysis is limited to ViT models. With P-IBISA, we visually show that other architectures, such as the ResNET-50, trained in an SSL fashion, can also learn semantically relevant features.

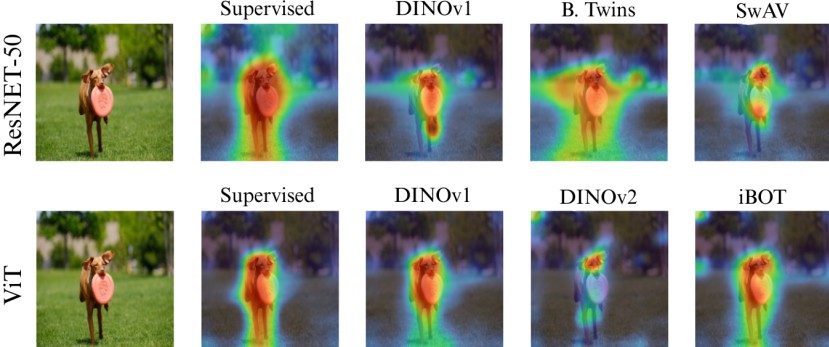

Figure 2: Saliencies generated to explain the encoder of the ResNET-50 and ViT models when trained with different strategies. For the supervised case, we have models trained for classification on the ImageNet dataset, while other cases are handled by models that are only pre-trained on unlabeled data. We see that all the attributions highlight regions that correspond to the target object. We use the ViT-B/16 in all our experiments, except for DINOv2, for which the weights are only available for the ViT-B/14 architecture.

## 6.1 NUMERICAL EVALUATION FOR IMAGE CLASSIFICATION MODELS

Table 1 presents the MoRF and LeRF metrics computed for ViT-B/16, VGG-16, and ResNET-50. Despite being model- and decoder-agnostic, our method demonstrates competitive performance compared to full guided approaches that rely on the model's final prediction, achieving the best or second-best results in half of the experiments. This indicates that the model's final output can be explained solely based on the encoder's behavior. A brief qualitative discussion regarding the decoder-agnostic explanations and the model's final prediction is presented on Appendix B.

Table 1: Metrics computed for the ViT-B/16, VGG-16, and ResNET-50 models on 10,000 images from the ImageNet validation set. Results presented for the scenario where $\beta$ and $\phi$ are learned during optimization. The Appendix C presents the results for fixed values of the hyperparameters.

| | ViT-B/16 | |
| --- | --- | --- |
| | MoRF ($\downarrow$) | LeRF ($\uparrow$) |
| Att R. | 0.218 | 0.643 |
| AG CAM | **0.190** | *0.645* |
| LRP based | *0.215* | 0.619 |
| P-IBISA | 0.217 | **0.676** |

| | VGG-16 | | ResNET-50 | |
| --- | --- | --- | --- | --- |
| | MoRF ($\downarrow$) | LeRF ($\uparrow$) | MoRF ($\downarrow$) | LeRF ($\uparrow$) |
| GradCAM | 0.079 | 0.546 | 0.338 | 0.637 |
| RISE | 0.082 | 0.601 | 0.296 | **0.687** |
| IBA | **0.077** | 0.594 | 0.286 | 0.665 |
| IBISA | *0.078* | **0.621** | **0.233** | *0.679* |
| P-IBISA | 0.079 | *0.616* | *0.254* | 0.674 |

## 6.2 USE CASE FOR VISUAL-LANGUAGE MODELS

P-IBISA establishes a new SOTA in explaining vision-language models. Figure 3 shows how our method can highlight regions of the image based on a text description. The text inputs indicated above each image are provided by BLIP (Li et al., 2022). These phrases are then tokenized and

passed through CLIP's text encoder, with the resulting embedding used as the target to compute the pairwise explanation.

'a little girl holding an umbrella in the rain'

'a horse pulling a carriage down a street'

'a man riding a wave on a surfboard'

'a small kitten sitting on a wooden bench'

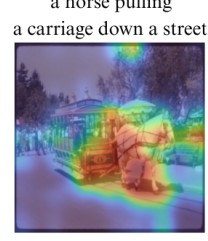 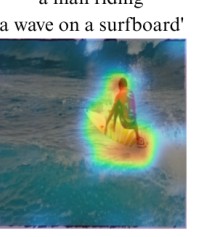 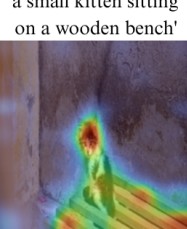

Figure 3: Images taken from the MS-COCO 2017 validation set. Saliencies generated using $\beta = 2$ and $\phi = 1$. Attributions generated for other values for the hyperparameters, including the case where they are learned during the optimization process, are found in the Appendix C.

Table 2 presents the CI and CD metrics computed for the text-image alignment task. We see that our method outperforms the current SOTA by a large margin on all datasets. Table 3 shows that our method achieves the best performance in the Cap MoRF and Cap LeRF metrics, and that all methods surpass the random baseline, indicating that these visual-textual explanations are indeed relevant. We also observe that the random baseline reaches similar metrics for Cap MoRF and Cap LeRF, which is expected since the importance of the pixels is determined randomly.

Table 2: Confidence increase and confidence drop results for all tested datasets and models. The results presented in the table are for saliencies generated for $\beta = 2$ and $\phi = 1$; results for other hyper-parameter values, including the case where they are learned during the optimization process, are found in Appendix C.

|  | MS-CXR | | Flickr8k | | MS-COCO | |
| --- | --- | --- | --- | --- | --- | --- |
|  | C. Drop ($\downarrow$) | C. Inc. ($\uparrow$) | C. Drop ($\downarrow$) | C. Inc. ($\uparrow$) | C. Drop ($\downarrow$) | C. Inc. ($\uparrow$) |
| RISE | 0.935 | 34.259 | *1.708* | 19.503 | *1.611* | 21.740 |
| M2IB | 1.231 | 36.111 | 2.017 | 18.107 | 1.764 | *23.800* |
| NIB | *0.642* | *52.778* | 1.947 | 18.712 | 1.657 | 23.500 |
| P-IBISA | **0.300** | **64.815** | **0.890** | **49.277** | **0.821** | **53.620** |

Table 3: Cap MoRF and Cap LeRF metrics for the saliency maps generated for CLIP. We observe that the random masks obtain similar results for Cap MoRF and Cap LeRF, which is expected, since the ordering of these pixels is arbitrary. We evaluated the METEOR (M), BLEU (B), ROUGE1 (R1), ROUGE2 (R2), and ROUGEL (RL) scores. The results presented in the table are for saliencies generated for $\beta = 2$ and $\phi = 1$; results for other hyper-parameter values, including the case where they are learned during the optimization process, are found in Appendix C.

|  | Cap MoRF ($\downarrow$) | | | | | Cap LeRF ($\uparrow$) | | | | |
| --- | --- | --- | --- | --- | --- | --- | --- | --- | --- | --- |
|  | M | B | R1 | R2 | RL | M | B | R1 | R2 | RL |
| P-IBISA | **0.334** | **0.154** | **0.389** | **0.253** | **0.363** | **0.651** | **0.482** | **0.683** | **0.593** | **0.663** |
| M2IB | *0.337* | *0.157* | *0.395* | *0.256* | *0.369* | *0.629* | *0.461* | *0.662* | *0.568* | *0.641* |
| NIB | 0.398 | 0.206 | 0.450 | 0.317 | 0.425 | 0.568 | 0.383 | 0.607 | 0.499 | 0.583 |
| Random | 0.493 | 0.307 | 0.534 | 0.418 | 0.511 | 0.494 | 0.308 | 0.535 | 0.419 | 0.510 |

## 6.3 IMAGE RETRIEVAL

Owing to its decoder-independent paradigm, image retrieval represents another well-suited application for our P-IBISA method, as we naturally focus on the encoder output and compare the generated embeddings rather than relying on a decoder component. In this context, P-IBISA facilitates the interpretation of pairwise similarities between images within the embedding space, effectively addressing the question: *"Based on which regions of the image did the model determine these images to be similar?"*

As shown in Figure 4, P-IBISA demonstrates more coherent patterns in areas indicating similarity between each retrieved image and its associated query, whereas the other xAI, such as Saliency Map (SBSM) (Dong et al., 2019) techniques, display weaker similarity relevance and greater sensitivity to noise. Applying P-IBISA provides an interpretable and clinically grounded indication of the extent to which the retrieval model emphasizes features that align with expert judgment when determining similarity, thereby enabling clinicians and researchers to identify problems within the developed retrieval framework effectively. For instance, the similarity between the retrieved B image and the query image is higher in the region corresponding to the watch that both subjects are wearing, which is undesirable for a model trained for this application.

In terms of metrics, as presented in Table 4, we observe that P-IBISA consistently surpasses the Integrated Gradients method (Sundararajan et al., 2017), and only falls second to SBSM in one case. These results highlight P-IBISA's capabilities for the retrieval task, presenting itself as the competitive choice for this application.

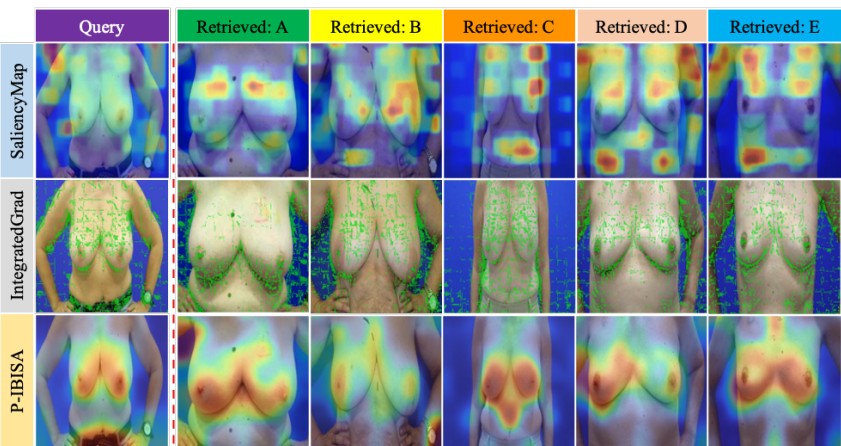

Figure 4: Since P-IBISA method only depends on the outputs of an encoder, its capabilities can be easily extended to image retrieval models. Saliencies generated using $\beta = \phi = 1$ and the mean squared error as the alignment loss. The saliency map on the query is the result of averaging the attributions generated on the query, for each retrieved image as the target.

Table 4: Confidence increase and confidence drop results for retrieval (Inverse Euclidean Distance).

| Method | Query | | Retrieved | |
|---|---|---|---|---|
| | C. Drop ($\downarrow$) | C. Inc. ($\uparrow$) | C. Drop ($\downarrow$) | C. Inc. ($\uparrow$) |
| Integrated Gradients | 4.258 | 3.511 | 4.186 | 5.143 |
| Saliency Map (SBSM) | 2.571 | **10.480** | 3.731 | 6.097 |
| P-IBISA | **1.996** | *5.568* | **1.461** | **29.270** |

## 7 CONCLUSION

This work introduced P-IBISA, a new model-agnostic xAI method to generate attribution maps for various applications, such as image classification, visual-language grounding, and image retrieval. Despite its model-agnosticism, we showed that P-IBISA achieves competitive performance on all of these tasks and sets the new SOTA for explaining visual-language grounding, outperforming the current methods by a large margin. P-IBISA also reveals that models trained with only unlabeled data learn human-relatable features, which was only known for ViTs, but P-IBISA also allows us to state this for CNNs.

As a limitation, we address that P-IBISA cannot generate a saliency for the text input in visual-language grounding, which M2IB and NIB are capable of. We also highlight that this limitation comes from the model-agnostic nature of P-IBISA. Although we can place a mask on top of a

phrase embedding and optimize it as we do for images, generating an attribution for each word in the phrase using only this information is not possible. We would need to access the tokenizer intermediate outputs to obtain such saliencies, thereby forfeiting the model-agnostic advantages of P-IBISA. We suggest this variation of our method as a potential direction for future work.

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

## A    CONTRASTIVE EXPLANATIONS

P-IBISA is also suitable for contrastive explanations, such as identifying regions that differentiate $f(x \odot m)$ from $h_t$. For this, the cosine embedding loss is now used to promote dis-alignment between the embeddings:

$$\mathcal{L}_{\text{dis-align}}(f(x \odot m), h_t) = \max\left(0, \cos\langle f(x \odot m), h_t\rangle\right) \tag{9}$$

By replacing $\mathcal{L}_{\text{align}}$ with $\mathcal{L}_{\text{dis-align}}$ in Eq. 6, the optimization now promotes the dis-alignment objective. Subtracting the saliency maps generated to explain *"Why similar?"* from the one generated to explain *"Why different?"* results in a contrastive attribution map for a given pair of images, as shown in Figure 5.

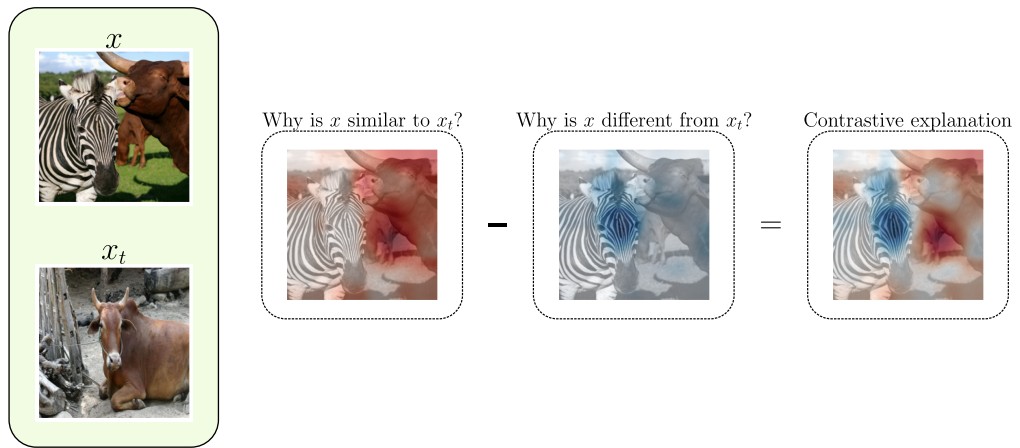

Figure 5: Beyond highlighting what regions are relevant to encode an image for its predicted embedding, our method can also find what makes two images similar or dissimilar. We compute the so-called contrastive explanation by subtracting the positive saliency from the negative one, with the blue regions representing what is most different and the red regions what is most similar. The saliency maps were obtained for the ResNET-50 model trained on ImageNet for classification.

## B    ENCODER EXPLANATION VS FINAL PREDICTION

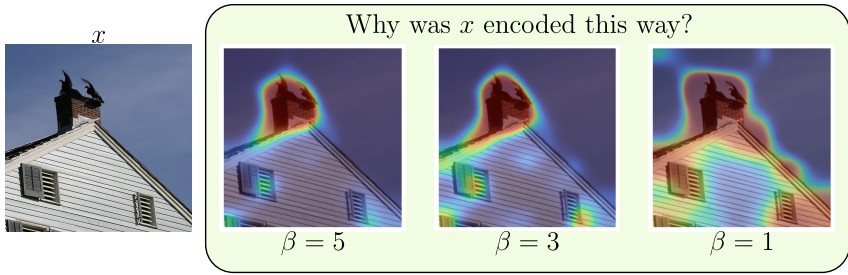

Figure 6: Results of P-IBISA to explain the encoder of a VGG-16 model trained on ImageNet. We noticed that the most important regions for the model's encoder are related to the vultures on the top of the building. On the other hand, the class predicted by the classification head of the network is *Church*. In this experiment, we used $\phi = 1$.

Observing Figure 6, one can notice that the vultures on the top of the building represent the most important region for the VGG-16 encoder. On the other hand, the top ten predictions for this model are, in order: church, beacon, picket fence, bell cote, tile roof, flagpole, dome, *vulture*, barn, solar dish. This indicates that the most relevant regions for the encoder do not necessarily correspond

to the final prediction of the model. Relaxing the bottleneck by reducing the value of $\beta$ reveals the regions the classifier head may have used to compute its most confident predictions. This type of analysis can be beneficial when debugging models that utilize transfer learning with a frozen encoder, as it allows one to evaluate whether the extracted features are semantically important for the downstream task beforehand.

## C NUMERICAL EVALUATION FOR FIXED PARAMETERS

Table 5: Metrics computed for the VGG-16, ResNET-50 and ViT-B/16 models on 10,000 images from ImageNET validation set.

| $\beta/\phi$ | VGG-16 | | ResNET-50 | | ViT-B/16 | |
|---|---|---|---|---|---|---|
| | MoRF ($\downarrow$) | LeRF ($\uparrow$) | MoRF ($\downarrow$) | LeRF ($\uparrow$) | MoRF ($\downarrow$) | LeRF ($\uparrow$) |
| Learnable | *0.079* | **0.616** | *0.254* | 0.674 | *0.217* | *0.676* |
| $\beta = 1/\phi = 1$ | **0.078** | *0.608* | 0.275 | 0.664 | 0.220 | 0.673 |
| $\beta = 10/\phi = 1$ | 0.086 | 0.596 | **0.227** | **0.683** | **0.213** | **0.679** |
| $\beta = 100/\phi = 1$ | 0.122 | 0.406 | 0.296 | *0.678* | 0.266 | 0.629 |

Table 6: Results for the Confidence Increase and Confidence Drop metrics using learnable and fixed hyper-parameters.

| $\beta/\phi$ | MS-CXR | | Flickr8k | | MS-COCO | |
|---|---|---|---|---|---|---|
| | C. Drop ($\downarrow$) | C. Inc. ($\uparrow$) | C. Drop ($\downarrow$) | C. Inc. ($\uparrow$) | C. Drop ($\downarrow$) | C. Inc. ($\uparrow$) |
| Learnable | *0.300* | *64.815* | 1.039 | 44.383 | 0.942 | 48.440 |
| $\beta = 1/\phi = 1$ | **0.139** | **77.778** | **0.382** | **67.680** | *0.948* | *48.800* |
| $\beta = 2/\phi = 1$ | 0.439 | 61.574 | *0.885* | *49.746* | **0.839** | **53.000** |
| $\beta = 3/\phi = 1$ | 1.308 | 37.500 | 1.769 | 31.998 | 0.945 | 48.280 |

Table 7: Results for the Cap MoRF and Cap LeRF metrics using learnable and fixed hyper-parameters.

| $\beta/\phi$ | Cap MoRF ($\downarrow$) | | | | | Cap LeRF ($\uparrow$) | | | | |
|---|---|---|---|---|---|---|---|---|---|---|
| | M | B | R1 | R2 | RL | M | B | R1 | R2 | RL |
| Learnable | 0.353 | 0.168 | 0.407 | 0.269 | 0.380 | 0.640 | 0.472 | 0.673 | 0.582 | 0.653 |
| $\beta = 1/\phi = 1$ | *0.341* | **0.154** | *0.397* | *0.259* | *0.370* | 0.645 | *0.477* | *0.680* | *0.588* | *0.659* |
| $\beta = 2/\phi = 1$ | **0.334** | **0.154** | **0.389** | **0.253** | **0.363** | **0.651** | **0.482** | **0.683** | **0.593** | **0.663** |
| $\beta = 3/\phi = 1$ | 0.346 | 0.165 | 0.399 | 0.266 | 0.374 | *0.646* | 0.474 | 0.679 | 0.587 | 0.658 |

# D SALIENCY EXAMPLES

## D.1 ENCODER EXPLANATION

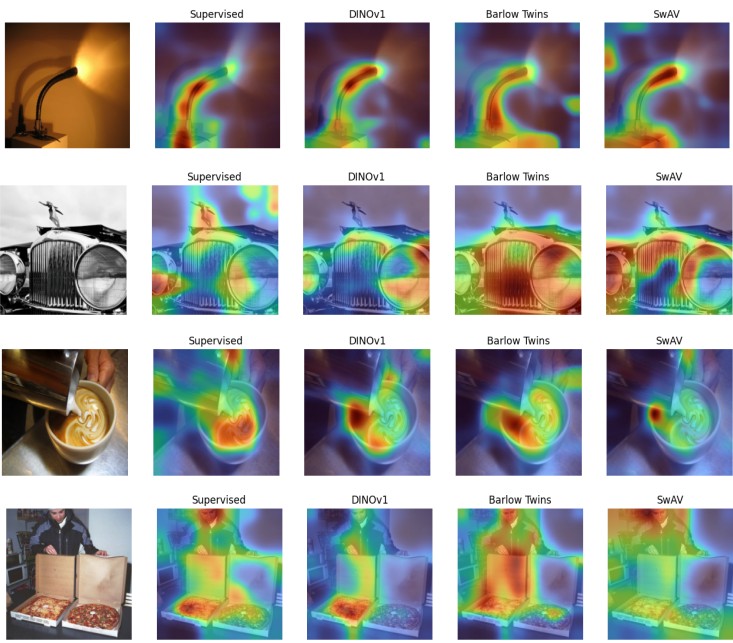

Figure 7: Explanations generated for the encoder of a ResNET-50 architecture trained with different learning strategies.

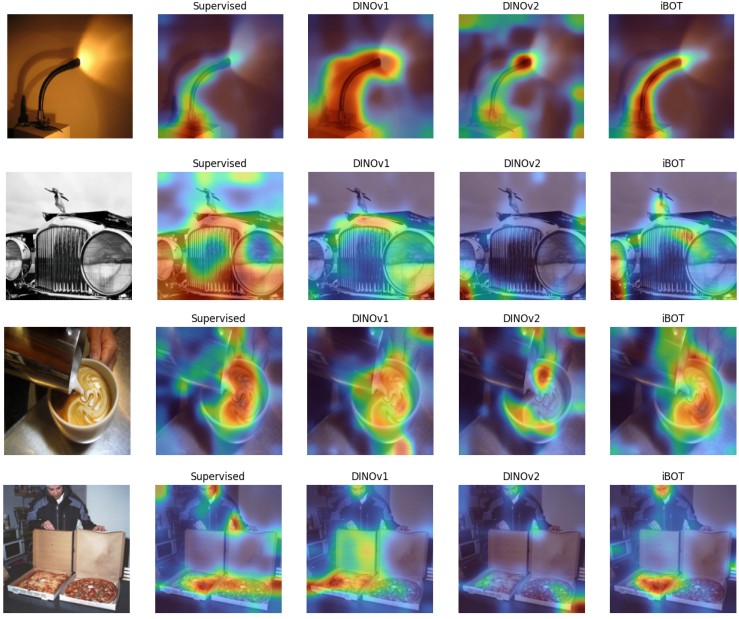

Figure 8: Explanations generated for the encoder of a ViT architecture trained with different learning strategies. We use the ViT-B/16 in all our experiments, except for DINOv2, for which the weights are only available for the ViT-B/14 model.

## D.2 Visual-language grounding

a red and black train is coming down the tracks

two cows grazing in a field with trees in the background

a bunch of already peeled oranges grouped together

a train traveling down train tracks near a train station

a person sitting down with a tennis racket

not really a good choice for this shirt and tie combination

Figure 9: Saliencies generated when explaining CLIP. We compare with the other SOTA methods NIB and M2IB. The prompt used to compute the explanations is located below each figure.

## E  Controlling the final attribution

Figure 10 presents the behavior of the saliency maps as a function of $\beta$ for all the studied models. We see that increasing $\beta$, while keeping $\phi$ fixed, results in more sparse attributions. One should also notice that choosing a value too high for $\beta$ ends up deteriorating the explanation (the case for the VGG-16 model), which is expected, since in the term that promotes sparsity in Eq. 6 becomes dominant over the others. For comparison, we also present the results for the case where $\beta$ and $\phi$ are learned during the optimization using Eq. 8.

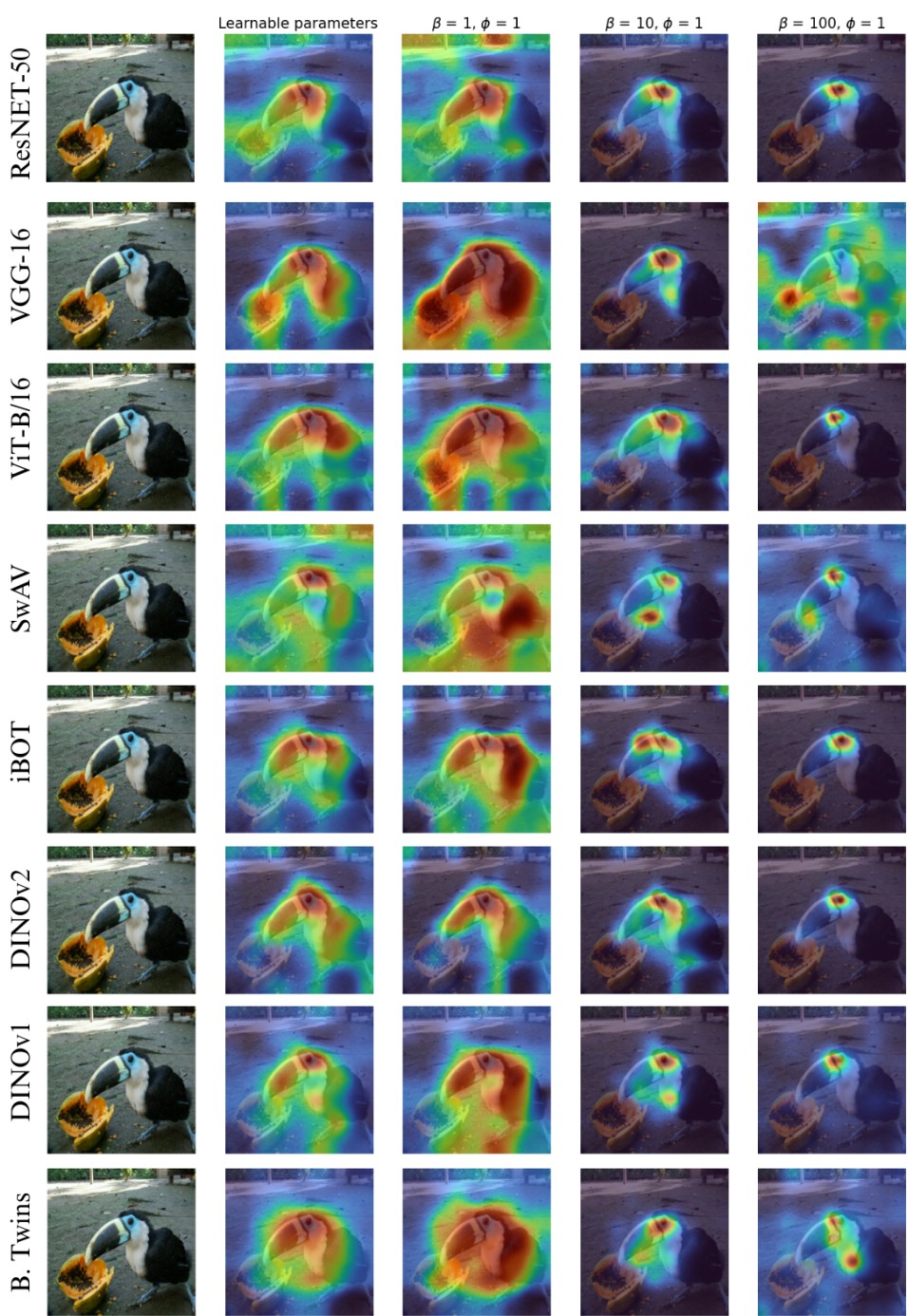

Figure 10: Explanations as a function of the $\beta$ parameter. In the image, the rows labeled as ResNET-50, VGG-16 and ViT-B/16 represent these models trained on a fully-supervised manner. The backbone of iBOT and DINOv2 are vision transformers, while for DINOv1, Barlow Twins and SwAV use the ResNET-50 as backbone.

