# OpenReview forum: "Pairwise explanations: towards a new task-agnostic paradigm in explainable artificial intelligence"
_ICLR.cc/2026/Conference — Submitted to ICLR 2026_

### Official Review · Reviewer_M1SE · 2025-10-28

**Soundness:** 3
**Presentation:** 3
**Contribution:** 3
**Rating:** 6
**Confidence:** 4

**Summary:**

The paper proposes P-IBISA (Pairwise-Information Bottleneck with Input Sampling for Attribution), a task-agnostic explainability framework that extends the IBISA method.
Unlike prior XAI techniques relying on task-specific outputs (e.g., logits), P-IBISA produces saliency maps directly from encoder embeddings, enabling explanations for classification, self-supervised, and vision–language models.
It optimizes differentiable input masks to align masked embeddings with a target embedding, balancing sparsity and smoothness through parameters (β, ϕ).
Experiments on ImageNet, MS-COCO, Flickr8k, MS-CXR, and a private medical dataset show competitive or state-of-the-art performance, especially in visual-language grounding.

**Strengths:**

- The idea of generating explanations entirely in the embedding space, independent of output logits, broadens interpretability to non-classification and multimodal tasks.

- The method is tested on classification, retrieval, and visual-language grounding tasks with both CNN and ViT architectures, demonstrating generality.

- Quantitative gains (especially for CLIP) and qualitative saliency maps show sharper, semantically coherent explanations than prior methods.

**Weaknesses:**

- The link between the modified loss (Eq. 8) and the Information Bottleneck principle is intuitive but not formally justified; claims about negative losses and stability remain heuristic.

- Improvements lack statistical validation; comparisons may favor P-IBISA due to differing hyperparameter schemes (learnable vs. fixed). Human or perceptual validation is absent.

- The approach still requires a differentiable encoder, so “task-agnostic” is limited to models with accessible gradients; text-side explanations are unsupported.

- Although the paper claims low computational cost, no runtime comparisons are presented. Sampling multiple masks and optimizing via Adam could be nontrivial.

**Questions:**

- How faithful are embedding-space explanations to model decisions, especially when the task head introduces non-linear transformations? Could P-IBISA highlight features irrelevant to the downstream output?

- What is the actual cost of optimizing multiple masks vs. sampling-based methods like RISE? Are there convergence or stability issues when learning β, ϕ?

- Could P-IBISA generalize to non-visual encoders (e.g., language or audio models)?

---

> ### Author Response · Authors · 2025-11-21
> **Responses to reviewer M1SE**
>
> >  WEAKNESSES
> 1. We agree with the reviewer that empirical findings mainly ground the formulation of the information bottleneck principle in P-IBISA. Nevertheless, the use of cosine similarity and mask complexity as a loss function achieves the information bottleneck objective in this context, as both are employed in the NIB and IBISA methods. A more formal definition of the problem would undoubtedly be very valuable for the work, but optimizing the mutual information directly requires it to be defined in a differentiable way. A workaround is to define an approximation for the mutual information that relies on assumptions about the probability distribution of the involved variables, assumptions that may not be feasible in the image input space, a requirement for P-IBISA.
> 2. As stated in the paper, negative loss functions have been applied in the literature and do not pose any theoretical problem for the gradient descent method. Positive loss functions are used to provide a readily interpretable measure of how well the models fit a ground truth distribution. In P-IBISA, the objective is to constrain the mask as much as possible while preserving the similarity between two embeddings, where there is no clear benefit or restriction to the use of a positive loss function.
> 3. Although statistical significance tests were not performed, we only claim that the P-IBISA sets the SOTA in the visual-language grounding task, where we surpass M2IB and NIB by a large margin in the established metrics for this task  (Confidence Increase and Confidence Drop). On the other metrics, we only claim that we achieve comparable performance with other popular methods, supporting the value of P-IBISA in being easily adapted to different tasks due to its decode-agnostic nature. For these reasons, even though statistical significance tests were not performed, our results and claims remain reasonable when considering only the absolute results for the metrics. Furthermore, the evaluation process conducted in this study aligns with standard practices in the machine learning field that were used in the M2IB and NIB works.
> 4. The literature may not have a consensus on the definition of model-agnostic. We consider P-IBISA a model-agnostic approach because its implementation does not change depending on the model being explained, as the entire optimization is performed in the input space. Nevertheless, P-IBISA is still a white-box method, as back-propagating through the network is required. RISE would be considered a model-agnostic black-box method because back-propagating through the network is not needed .
>
> > QUESTIONS
>
> 1. We evaluated the faithfulness of the embedding space when assessing the performance of P-IBISA in the classification task, where a saliency map computed only with the encoder was found to be relevant in explaining the model's final prediction, as evidenced by the MoRF and LeRF metrics. Nevertheless, it is indeed possible for features not relevant to the downstream task to be highlighted. We included a brief discussion about this in the Appendix B of the revised manuscript, where we analyse how the regions highlighted by P-IBISA are related to the predictions of a classification model.
> 2. When compared to other model-agnostic methods, we can consider P-IBISA a computationally efficient approach, given that it has the exact cost of IBISA, which was shown in the original article to be at least $8 \times$ more time efficient than RISE. While P-IBISA optimizes 20 masks for 15 steps, resulting in 300 forward passes and 300 backward passes, RISE typically requires between 5,000 and 6,000 forward passes to compute a single saliency map, clearly indicating improved efficiency. The learning procedure of the $\beta$ and $\phi$ parameters is well established in the literature, and we did not encounter any issues regarding this in our experiments.
> 3. Generalization to other domains is a step to be considered in future work. A priori, we could optimize the mask over inputs of any shape; however, further experiments are necessary to support this hypothesis.

---

> > ### Comment · Reviewer_M1SE · 2025-11-26
> >
> > Thanks for the clarification and the rebuttal. The authors provide helpful explanations regarding the distinction between white-box and model-agnostic approaches, the computational efficiency relative to RISE, and the practical challenges of directly optimizing mutual information in image space. These clarifications improve my understanding of the design choices behind P-IBISA.
> >
> > However, the rebuttal only partially addresses some of the concerns. The theoretical grounding of the loss formulation remains heuristic, and referencing similar assumptions in prior work does not fully establish a principled connection to the Information Bottleneck objective. In addition, there was no meaningful clarification of the claims surrounding task-agnosticism.
> >
> > Overall, the rebuttal improves clarity but does not fully resolve the deeper methodological and empirical issues raised in the initial review.

---

### Official Review · Reviewer_ZtuP · 2025-11-01

**Soundness:** 2
**Presentation:** 3
**Contribution:** 3
**Rating:** 4
**Confidence:** 5

**Summary:**

The paper proposed an explanation method for black-box models which generates a saliency map to localize the important features on the instance image. The basic idea is to find a coupled image for the instance adopt the information bottleneck criteria to extract the important embedding features that share between both images. The authors also extend the method to achieve image-to-text explanations.

**Strengths:**

The authors combined previous metrics adopted in information bottleneck explanation methods which reached a more comprehensive scoring function that may be potentially advantageous.

The idea of adopting IB to achieve cross-modal alignment is interesting.

**Weaknesses:**

The method requires a real sample image with known mask (saliency) to serve as the target for alignment, which poses two limitations. Firstly, the initial inaccuracy for the target mask will hinder the outcome. Secondly, the background variances between samples will degrade the accuracy.

The experiment part is flawed. Firstly, the method is based on IBISA, so IBISA should be compared for ablation study. Secondly, the authors should not compare only information bottleneck methods, but should also include SOTA xAI methods of other categories that can also generate a saliency map, such as counterfactual generation methods which also adopt paired comparisons and are more powerful in tackling background variances.

**Questions:**

The initial choice of paired image and generation of target masks are crucial for the results but was not described at all. Could you elaborate both of them?

---

> ### Author Response · Authors · 2025-11-21
> **Response to reviewer ZtuP**
>
> >WEAKNESSES
> 1. It is worth noting that our method does not rely on a real image with a known mask. Given a target image $x_t \in \Re^{3 \times W \times H}$, we obtain a target embedding $h_t = f(x_t)$, with $f(\cdot)$ being the image encoder and $h_t \in \Re^{1 \times D}$. We then find a mask $m$ that, when applied over another image $x_i$, approximates the two embeddings $h_t$ and $h_i = f(x_i \odot m)$; a problem that is intuitively formulated using the Information Bottleneck Principle. For visual-language models, the target embedding is obtained by passing a phrase through the text encoder, and the optimization to obtain the saliency is conducted in the same manner.
> 2. Since IBISA is an xAI method for classification models, we compared it with P-IBISA only on the classification task. One should note that P-IBISA is not a method designed explicitly for classification, as the classification head is not used to generate explanations. Nevertheless, we still obtained reasonable performance compared to other methods in this task, which also included gradient and perturbation-based methods. On the visual-language grounding, we compared with the SOTA methods M2IB and NIB that are also based on the Information Bottleneck principle, where we beat the SOTA by a large margin in the Confidence Increase and Confidence Drop metrics.
>
> > QUESTIONS
> - As described before, we do not rely on a target (ground truth) mask. A target image is only necessary for the retrieval task, which is the image that the model retrieves. For the classification task, the target image is the image itself, and for the visual grounding, the target embedding is obtained from a textual prompt. We agree with the reviewer that this information could be better presented and clarified; therefore, we have updated Figure 1 of the manuscript to provide a more precise explanation of this process.

---

> > ### Comment · Reviewer_ZtuP · 2025-11-26
> >
> > Thank the authors for clarification. However, I didn't mean that you would need a ground-truth mask. Instead I have talked about the  "generation" of the mask (which the authors have answered) and the "inaccuracy" of the mask (which I think remains a major problem).
> > Also, I insist that the comparisons should not be restricted to IB-based approaches, because users don't care what techniques you used to generate explanations.

---

> > > ### Author Response · Authors · 2025-12-02
> > > **Response to Reviewer ZtuP**
> > >
> > > We thank the reviewer for the comments. We disagree that the "inaccurate" mask generation process is a limitation of our work. Due to the random initialization, the masks converge to different local minima at each iteration, which is beneficial for our method, as it enables us to identify different important regions in the image. These optimized masks are then aggregated to compute the final saliency, which, in our experiments, did not pose any problem regarding stability and allowed P-IBISA to reach SOTA results in the visual-language grounding task. Beyond that, input perturbation methods are common in the literature of xAI for attribution generation; methods such as RISE and IBISA rely on similar strategies.
> > >
> > > We also want to clarify that our comparisons are not limited to IB-based approaches. We also compared our method against RISE, Grad-CAM, Attention Rollout, LRP, Integrated gradients, SBSM, which are not based on the IB theory. On the visual-language grounding, the comparison is limited to M2IB and NIB because these are the SOTA methods for the task, as shown in the previous edition of this conference in the NIB paper [Zhu, et al. 2025 ](https://openreview.net/forum?id=INqLJwqUmc).

---

### Official Review · Reviewer_sx2L · 2025-11-01

**Soundness:** 2
**Presentation:** 2
**Contribution:** 2
**Rating:** 2
**Confidence:** 4

**Summary:**

This paper introduces P-IBISA (Pairwise Information Bottleneck with Input Sampling for Attribution), a novel model-agnostic eXplainable AI (XAI) method that extends the Information Bottleneck Principle (IBP) to generate saliency maps. Unlike traditional attribution methods that rely on task-specific logits, P-IBISA works directly on the encoder's embedding space by optimizing a mask over the input image to align its masked embedding with a target embedding (which can be derived from another image, text, or a different task model). The method is positioned as a task-agnostic framework capable of providing explanations across image classification, visual-language grounding (e.g., CLIP), and image retrieval tasks.

**Strengths:**

1. P-IBISA successfully demonstrates its ability to generate saliency maps for various models (CNNs and ViTs) and tasks (classification, VLM grounding, retrieval) using a unified, model-agnostic approach based purely on the encoder's embedding.

2. The method achieves state-of-the-art performance on the quantitative Confidence Drop (CD) metric across all three evaluated tasks (MS-CXR, Flickr8k, MS-COCO), suggesting the generated saliency maps effectively capture the most crucial predictive regions.

3. The generated saliency maps, particularly in the image retrieval and visual grounding tasks, appear visually coherent and semantically relevant to the paired concepts, supporting the claim of generating "pairwise explanations."

**Weaknesses:**

1. The core weakness is the conceptual ambiguity of the "task-agnostic" claim. While the method uses the encoder embedding (which is "task-agnostic" to the classifier head), the target embedding ($h_t$) must still be provided by a task-specific mechanism (e.g., a text encoder for VLM, another image's encoder for retrieval). This dependency means the method is only decoder-agnostic, not truly task-agnostic, and the complexity introduced is not fully justified over simpler model-specific techniques.

2. The method is highly complex, involving mask optimization, two sparsity and smoothness regularizers ($\beta$ and $\phi$), and the need to optimize masks over multiple random initializations and average them. This significant computational overhead and complexity are not adequately justified by the incremental performance gains over existing, simpler model-agnostic baselines like Integrated Gradients or Saliency Map.

3. The paper claims a significant advantage over methods like M2IB and NIB because those methods cannot generate explanations for text inputs. However, this is an unfair comparison since those methods were not designed for multimodal tasks. The appropriate comparison should focus on whether P-IBISA is the best XAI approach for the image portion of the VLM task, which is not clearly demonstrated.

4. The reliance on multiple random mask initializations to ensure robustness (Sec 3.1) suggests a fundamental stability issue in the core optimization problem. A more rigorous analysis of why the single-initialization optimization fails or why this averaging is necessary is missing. Moreover, the lack of an ablation study on the $\beta$ and $\phi$ hyperparameters, which control crucial sparsity and smoothness, leaves the method highly vulnerable to practitioner use.

**Questions:**

1. Please clarify the computational cost of P-IBISA relative to simpler gradient-based methods. Given that P-IBISA requires optimizing the mask $m$ for multiple random initializations (Equation 2) and uses an iterative optimization loop (Algorithm 1), how does the total runtime (per explanation) compare to baselines like Integrated Gradients?

2. The "task-agnostic" claim is highly ambitious. Could the authors refine this claim to "decoder-agnostic" and discuss the practical limitations that arise from the dependency on a pre-calculated target embedding ($h_t$)? Specifically, how is $h_t$ obtained in a truly novel, unseen task setting?

3. The paper uses a weighted combination of three terms in the total loss $\mathcal{L}_{\text{P-IBISA}}$ (Equation 6). An ablation study on the loss terms and the $\beta$ and $\phi$ hyperparameters is crucial. What is the impact of varying the coefficients of the sparsity/smoothness terms on the MoRF and CD metrics?

---

> ### Author Response · Authors · 2025-11-21
> **Response to Reviewer sx2L**
>
> WEAKNESSES
> 1. We agree that positioning our method as a decoder-agnostic approach, which leads towards a task-agnostic approach, is more suitable based on the results we present. This new positioning is now reflected in the revised manuscript. Since any model is optimized for a given task, even when using self-supervised strategies, we did not understand what the reviewer meant by "*the target embedding () must still be provided by a task-specific mechanism*". What would be a *non-task-specific mechanism*? Nevertheless, the task-specific claim suggests that the decoder independence makes it very easy to adapt P-IBISA for models trained for different tasks, as evaluated in the paper.
> 2. The optimization process for the masks of P-IBISA does not incur prohibited computational costs. P-IBISA is actually more computationally efficient than other SOTA agnostic methods, such as RISE, where at least 5000 forward passes are used to compute one attribution. In comparison, for P-IBISA, only 300 forward passes and 300 backward passes are needed. Compared to other methods, such as Integrated Gradients and SBSM, we incur a higher computational cost; however, the generated saliencies are considerably improved, as stated in Section 6.3 of the manuscript.
> 3. Regarding the third comment, we would like to clarify that M2IB and NIB are methods designed to explain CLIP, a visual-language model that is also capable of generating saliencies for text. On the other hand, to avoid compromising model-agnosticism, we do not compute saliencies for the text input, as it would be necessary to access intermediate layers of the model's text encoder, as the authors of NIB did to compute RISE over the text input.
> 4. We do not believe that the process of sampling different masks and averaging them to obtain the final attribution is a limitation of our work. Since each mask is initialized randomly, the final optimized mask depends on its initial state. Averaging a $k$-number of optimized masks is beneficial for providing more consistent attribution, a common procedure across many scientific domains. However in the work of IBISA, it has been shown that the Regarding the hyper-parameters, we refer the review to Appendix C, where the ablation studies are presented.
> > QUESTIONS
> 5. When using a Macbook Air with a M1 CPU and 16GB of RAM, without GPU acceleration, Integrated gradients takes 13.85 seconds to compute the attribution for an image pair, faster than SBSM (32.71 seconds) and P-IBISA (56.42 seconds). Nevertheless, the time required to compute P-IBISA is not prohibitive. The trade-off between computational cost and attribution quality pays off, as the saliencies presented in Figure 4 are more consistent than those of the other methods. We also achieved the best performance on all metrics except one.
> 6. We agree with the reviewer, and in the revised manuscript, we have positioned P-IBISA as a decoder-agnostic method, which is a step towards task-agnosticism. We do not consider the process of obtaining the target embedding a limitation, since we assume that it is possible to perform a forward pass through the network's encoder, regardless of its architecture or the downstream task for which the model is trained.
> 7. An ablation study was performed in the original manuscript, but due to the page limit, it is placed in Appendix C. We observed that learning the parameters during the optimization does not necessarily result in the best metrics, with some empirical fixed set of values for $\beta$ and $\phi$ being the best choice. On the other hand, having the possibility of learning the parameters reduces the workload on the end-user, without significantly compromising the method's performance. Appendix E also presents how the saliency changes with the choice of $\beta$.

---

### Official Review · Reviewer_mT9S · 2025-11-01

**Soundness:** 2
**Presentation:** 2
**Contribution:** 2
**Rating:** 2
**Confidence:** 4

**Summary:**

This paper proposes P-IBISA (Pairwise-IBISA), an extension of the Information Bottleneck with Input Sampling for Attribution (IBISA) framework, designed to generate task-agnostic explanations directly from encoder representations rather than output logits. The method aims to provide a unified framework for explainability across tasks such as image classification, vision–language grounding, and image retrieval.

**Strengths:**

The topic is timely and practically important, addressing the increasing need for unified, model-agnostic interpretability methods applicable across diverse architectures and tasks.

The proposed pairwise formulation introduces a flexible perspective that can potentially apply to tasks like image retrieval or cross-modal alignment.

**Weaknesses:**

The proposed approach mainly extends IBISA by substituting logits with encoder embeddings and introducing learnable regularization terms. This is a modest technical modification rather than a fundamentally new framework. The paper does not clearly articulate how P-IBISA advances the state of the art beyond prior information bottleneck–based and model-agnostic explanation techniques. The claimed “task-agnostic” property is somewhat overstated since the method still depends on task-specific embedding structures.

Many reported results are only marginally better or even comparable to baselines. In some cases, the choice of baselines omits more recent explainability methods, making the reported superiority claims unconvincing. The paper asserts consistent state-of-the-art performance, but quantitative results do not consistently demonstrate clear or statistically significant improvements over baselines.

The qualitative results presented are limited and largely anecdotal. The paper would benefit from more visual examples, including failure or ambiguous cases, to help readers understand the model’s strengths and limitations.

Several aspects of the method, such as the rationale for allowing negative loss values and the stability of learnable sparsity parameters, lack rigorous justification or ablation studies.

Key figures (especially Figure 1) are difficult to interpret, and the roles of different components (e.g., pair image, target embedding, masked image) are not clearly delineated.

The field of attribution and explainability is already well-explored, and without clearer differentiation or deeper insights, the paper’s contribution risks being perceived as incremental.

I found several typos throughout the paper. Please proofread.

**Questions:**

How does P-IBISA fundamentally differ from the original IBISA framework beyond replacing logits with encoder embeddings? Could you clearly articulate the theoretical or algorithmic innovation that makes this a new paradigm rather than an incremental variant?

Why were more recent or advanced xAI methods (e.g., LRP variants, concept-based explainers, or feature steering approaches) not included as baselines? Would incorporating these methods change the relative performance claims of P-IBISA?

Some experimental results show only marginal or inconsistent improvements. Have you performed statistical tests (e.g., significance or confidence intervals) to ensure the observed differences are meaningful and reproducible?

Can you include additional qualitative examples, especially failure or ambiguous cases, to better illustrate when P-IBISA succeeds or fails? This would provide readers with a more comprehensive understanding of the method’s behavior across contexts.

---

> ### Author Response · Authors · 2025-11-21
> **Response to reviewer mT9S**
>
> 1. We agree that the novelty of P-IBISA is incremental, but, to the best of the authors' knowledge, it is the first time attributions are generated independently of the model's final output. This change, although subtle, resulted in a method that advances the SOTA for attribution generation in visual-language models, a topic that has been gathering attention from the xAI community in recent years. Additionally, the model (decoder)- agnostic nature of P-IBISA makes it a flexible method that can be easily applied in different contexts, for different architectures.
> 2. P-IBISA sets the SOTA for visual-language grounding, where we surpass the most recent methods M2IB and NIB by a large margin in the Confidence Increase and Confidence Drop metrics, which were used in these previous works. For the classification task, our results show that P-IBISA achieves comparable results with other methods, indicating that an attribution computed with only the encoder is faithful to the model's final prediction. Since P-IBISA is easily applicable to different tasks, our goal when showcasing results for the classification task was not to show that P-IBISA surpass the SOTA, but to show it can also be applied to different tasks, which does not justify a broader comparison. The same applies to retrieval, where the pairwise nature of P-IBISA makes it suitable for the task. However, a broader evaluation of this task is necessary; thus, we only claim that it outperforms the other methods in our specific use case.
> 3. We thank the reviewer for the suggestion and we have now included more qualitative examples of the generated attributions in the new Appendix D of the revised manuscript.
> 4. As stated in the paper, negative loss functions have been applied in the literature and do not pose any theoretical problem for the gradient descent method. Positive loss functions are used to provide a readily interpretable measure of how well the models fit a ground truth distribution. In P-IBISA, the objective is to constrain the mask as much as possible while preserving the similarity between two embeddings, where there is no clear benefit or restriction to the use of a positive loss function. The process of learning the hyperparameters during the optimization process follows a well-established method in the literature, as referenced in the manuscript. Furthermore, Appendix C presents the ablation studies for the $\beta$ and $\phi$ parameters, comparing them when fixed to a given value versus when learned during optimization. We show that the metrics do not deteriorate in any of the cases, indicating stability of the explanations. Appendix E also presents how sparse the attributions can be based on the values of these hyperparameters. We clarified in the updated manuscript that these ablations are present in Appendix B.
> 5. Figure 1 was updated in the new version of the manuscript. This new version better clarifies our method.
> 6. We agree that the field is well-explored, but there are still gaps that need to be studied. Attribution methods for visual-language models, such as CLIP, are scarce in the literature, with the primary methods, M2IB and NIB, being published in NeurIPS 2023 and ICLR 2025, respectively. The same can be said of the retrieval task, where the leading solutions are adaptations of methods designed for classification that do not perform well on this task, such as the methods cited in the manuscript. P-IBISA emerges as a solution to fill some of these gaps, especially in the visual-language grounding task. We also emphasize the importance of a model (decode)-agnostic approach, which is valuable within the context of artificial intelligence, where new architectures and training objectives are proposed at a rapid pace.
> 7. We apologize for the typos in the original submission. The new version is now updated with the corrections.
> 8. We did not perform a statistical evaluation of our results, which would be beneficial for our work. On the other hand, we only claim that P-IBISA sets the SOTA in the visual-language grounding task, where we surpass M2IB and NIB by a large margin in the established metrics for this task  (Confidence Increase and Confidence Drop). On the other metrics, we only claim that we achieve comparable performance with other popular methods, supporting the value of P-IBISA in being easily adapted to different tasks due to its decoder-agnostic nature. For these reasons, even though statistical significance was not performed, we consider that our results and claims remain reasonable when considering only the absolute values for the metrics. Furthermore, the evaluation process conducted in this study aligns with the standard practices in the machine learning field and was used in the papers that introduced M2IB and NIB .

---

### Official Review · Reviewer_6qZz · 2025-11-05

**Soundness:** 2
**Presentation:** 2
**Contribution:** 2
**Rating:** 2
**Confidence:** 3

**Summary:**

This paper introduces Pairwise-IBISA (P-IBISA), an extension of the Information Bottleneck with Input Sampling for Attribution (IBISA). The method generates saliency maps directly from encoder representations instead of model outputs, allowing attribution across different tasks such as image classification, vision–language alignment, and image retrieval. The authors evaluate P-IBISA on several benchmarks and report competitive or state-of-the-art performance on quantitative metrics including MoRF, LeRF, Confidence Drop, and Confidence Increase.
The methodological contribution of moving from task-specific to encoder-based attribution is an interesting technical step and represents a modest but genuine improvement over previous work.

This paper presents an interesting technical extension of IBISA and shows credible quantitative improvements. However, it suffers from overstated interpretability claims, unclear qualitative evidence, confusing figures, and a lack of ethical transparency concerning the medical dataset. The work represents an incremental technical advance but does not substantiate its broader claims about explainability or human understanding.

**Strengths:**

-	The paper presents a coherent technical advancement by making IBISA encoder- and task-agnostic. This may be useful for understanding models where the output layer is not directly accessible.
-	The experiments are broad in scope and demonstrate the method across several domains, with consistent implementation and comparison to relevant baselines.
-	Quantitative results indicate good performance and reduced computational cost relative to some existing gradient-based approaches.

**Weaknesses:**

Overstated claims of explainability - The paper repeatedly overstates its contribution to explainable artificial intelligence. While P-IBISA provides a novel attribution mechanism, the authors make strong claims about improved human interpretability that are not supported by evidence. No user study or structured qualitative analysis is conducted. The abstract and Section 6 refer to a “qualitative analysis”, but the paper only includes example saliency maps without much evaluation.
The paper also states “P-IBISA also reveals that models trained with only unlabelled data learn human-relatable features” which is not particularly supported by evidence. The presented saliency maps only show that self-supervised models focus on object regions, which does not demonstrate human-relatable feature learning… This observation is already well established in self-supervised learning literature and does not result from the proposed explainability method.
Confusing and inconsistent figures - Several figures are unclear or contain errors. I.e.
Figure 2 has a caption that is grammatically incorrect and largely unintelligible (“we the are models… others cases are care for models…”), which makes interpretation difficult.
The contrastive explanation examples in Appendix A are confusing and seemingly counterintuitive – the same image is compared with itself showing only erroneous similarities and, more interestingly, strong contrast.
References to additional qualitative examples, such as those supposedly in Appendix B, do not correspond to any actual figures.
Overall, the visual presentation does not effectively support the claims about interpretability.
Missing or unclear supplementary materials (as mentioned above) – mentioned in the caption of figure 3.
Use of private medical data -
Section 6.3 introduces a private dataset of breast photographs used for a medical image retrieval experiment. This seems unmotivated and raises a few questions. Firstly, there is no ethics statement, institutional approval, or information about patient consent. Secondly, the dataset is private and therefore no reproducible. The inclusion of this material is not well motivated and does not clearly relate to the main technical focus of the paper (why not use a publicly available medical dataset?). Furthermore, the resulting saliency maps highlight the obvious breast regions, which provides no meaningful clinical insight. This choice seems rather out of place from the rest of the paper.

**Questions:**

N/A

---

> ### Author Response · Authors · 2025-11-21
> **Response to Reviewer 6qZz**
>
> We thank the reviewer for its comments regarding the strengths of our work. We also understand the reviewer's concerns and tried to address them the best as possible.
> > WEAKNESSES
> - We agree with the reviewer regarding some of the manuscript's weaknesses. We updated the manuscript to correct the typos and revised the figures to improve their readability. We also provide more qualitative examples. In Appendix B, we evaluate the correspondence between the decoder-agnostic explanation and the final prediction of a classification model. In Appendix D, we show more attribution examples and comparisons with the M2IB and NIB methods.
> - On the other hand, we respectfully believe the reviewer underestimates some of P-IBISA's contributions. The literature regarding saliency maps generation methods, although vastly explored, still has some gaps to be filled, such as the visual-language grounding application, where P-IBISA sets the new SOTA previously held by the NIB (published in the past edition of this conference), showing its meaningful contribution to the xAI community. The critique regarding the improved interpretability can be extended to any attribution method, not P-IBISA itself. Since computer vision models rely on processing local information in the input image, identifying which regions are most relevant to the model's decision is a faithful way to interpret its decision-making process. Visually indicating whether the region highlighted by the model corresponds to the object in the scene, along with the faithfulness metrics presented in the work, enhances human interpretability. In the benchmark datasets used in the paper, interpretation is readily assessable by the reader, as it is only related to common object images, and no specialized qualitative study is then required. On specialized domains, the relevant information in the input is only available to an expert practitioner, who can thus use attribution methods to sanity-check whether the developed models are learning human-relevant features. Nevertheless, we agree that a qualitative study with domain specialists is needed to further validate our method, and we leave it as future work.
> - When it comes to explaining the models trained with self-supervised strategies, we only highlighted that P-IBISA is a method capable of generating saliencies for models without a downstream task, something new for an attribution method. The findings in the SSL literature for ViTs about the relevance of the learned features can also be evaluated for CNN backbones through the insights provided by P-IBISA.
> - We agree with that the retrieval case in private medical dataset might seem unmotivated and thus we have clarified its importance in the revised manuscript. The retrieval application is presented in the work to illustrate the capabilities of P-IBISA in a specialized domain, specifically for a task where attribution methods have not been extensively explored. Again, the regions highlighted by P-IBISA improve interpretability. When developing a retrieval solution, it is important to debug if the obtained model is aligning the representation based on the relevant information for the task, in this case, the patient's breast, instead of spurious elements that are common for all the images, such as the background, accessories, belly, shoulders, arms, etc. In fact, observing the third column of Figure 4, one can see that the query patient and the retrieved B patient are both wearing a watch, which is pointed by P-IBISA as a region of high similarity between the two images, a behavior that not desired by this retrieval model.  This information may not be relevant to the clinician, but it is important for machine learning practitioners, something that the other tested methods failed to provide. Finally, the dataset was collected within the scope of a well-established project, adhering to all relevant ethical guidelines, and has been utilized in previous publications of the group. More details about the dataset were omitted to preserve the authors' anonymity; however, they will be available in the camera-ready version of the paper. Comments regarding these discussed topics are now present in Sections 5 and 6.3 of revised version of the manuscript.
> - We again thank the reviewer for its comments, and hope that we clarified most of the concerns regarding our manuscript.

---

### Author Response · Authors · 2025-11-21
**General comment for the reviewers**

We thank all the reviewers for their comments regarding our work. We are pleased that the main strengths of our paper are recognized, and we understand the concerns of the reviewers regarding the paper's presentation and details about the method's workings. We carefully considered all the comments and suggestions made by the reviewers and attempted to address all the issues pointed out to improve the quality of our work. In the revised version of the manuscript, we corrected typos and refined the overall text for improved readability. We also improved Figures 1 and 5, which now provide a better understanding of the P-IBISA method and the contrastive explanation paradigm. Finally, we have added Appendices B and D, which discuss the relationship between the decoder-agnostic explanation and the model's final prediction, and showcase additional qualitative examples and comparisons with other methods. We again thank the reviewers for the kind and informative evaluation of our work.

---

### Meta-Review · Area_Chair_7Nqy · 2026-01-04

**Summary:**

The paper was reviewed by 5 experts with scores 22624. The Reviewers' concerns are listed below.  The response did not address the numerous concerns in a convincing way, and mostly sidestepped the problems rather than try to provide further evidence. Overall, the paper is not ready for publication.

**Reviewer Concerns:**

**Reviewer 6qZz**
1. overstated claims not supported by evidence.
2. confusing/inconsistent figures
3. missing/unclear supplementary materials
4. use of private medical data, and no meaningful clinical insight given.

The AC thinks that the concerns were not addressed well, in particular the lack of user study (claim about improved human interpretability). The publication of images from private medical data also is concerning, and the AC wonders if the participants of that study had given permission to have their images published.

**Reviewer mT9S**
1. incremental novelty - extending IBISA.
2. marginally better results - needs statistical significance tests.
3. needs more visual examples / qualitative results.
4. needs more rigorous justification for some design aspects.
5. more recent XAI methods were not included as baselines.

The AC thinks that the concerns were not addressed very well. In particular the novelty aspect, and the need for statistical tests to show actually statistically significant improvement. In addition, the AC also notes that there are other attribution methods for CLIP:
- Gradient-based Visual Explanation for Transformer-based CLIP, ICML 2024.
- Grad-ECLIP: Gradient-based Visual and Textual Explanations for CLIP, https://arxiv.org/abs/2502.18816

**Reviewer sx2L**
1. "task-agnostic" is an over-claim.
2. significant computational overhead.
3. unfair comparison with M2IB and NIB, and should focus on image encoder.
4. stability issue due to random mask initializations? No ablation study.

The AC thinks that concern 2 was not addressed well -- sampling induces significant cost compared to gradient-based methods. Also Concern 4 could have been addressed better, e.g., using stability analysis of the results for different # of masks, or seeds.

**Reviewer ZtuP**
1. limitation due to sampling with a possibly inaccurate mask.
2. lacks ablation study using IBISA, and also missing comparison with non IB methods.

The AC thinks the concern 1 was not addressed well, since it was about the stability of the result due to random initializations. Concern 2 was also not addressed well, since there exist other methods that can be applied to CLIP.

**Reviewer M1SE**
1. only intuitive link between IB and modified loss (Eq 8).
2. needs statistical tests to validate improvements. missing user study.
3. "task-agnostic" is over claimed, limited to models with accessible gradients.
4. claim of low computational cost is not validated.

The AC thinks that Concerns 2, 3, 4 were not addressed well.

**Reviewer Scores:**

Reviewer 6qZz stated that they will maintain their rating (2), while Reviewer m1SE was not satisfied with the response, and likely would keep their rating (6).
Other reviews most likely would not increase their ratings beyond marginal reject, since there were still significant concerns that were not addressed well enough.

---

### Decision · Program_Chairs · 2026-01-26

Reject